# Photoelectrochemical Fe/Ni cocatalyzed C−C functionalization of alcohols

Long Zou[1,2], Rui Sun[1,2], Yongsheng Tao[1], Xiaofan Wang[1], Xinyue Zheng[1] & Qingquan Lu ●[1] ✉

The simultaneous activation of reactants on the anode and cathode via paired electrocatalysis has not been extensively demonstrated. This report presents a paired oxidative and reductive catalysis based on earth-abundant iron/nickel cocatalyzed C−C functionalization of ubiquitous alcohols. A variety of alcohols (i.e., primary, secondary, tertiary, or unstrained cyclic alcohols) can be activated at very low oxidation potential of (~0.30 V vs. Ag/AgCl) via photoelectrocatalysis coupled with versatile electrophiles. This reactivity yields a wide range of structurally diverse molecules with broad functional group compatibility (more than 50 examples).

Electrochemistry represents an environmentally benign method in organic synthesis because it does not require reactive (sometimes dangerous) chemical oxidants or reductants[1–13]. However, owing to the large interelectrode distance (from millimeters to centimeters), achieving selective cross-coupling of highly reactive intermediates via convergent paired electrolysis remains a significant challenge[14]. As a result, most reported electrosynthesis methods rely on single half-electrode reactions, where either H₂ evolution at the cathode or a sacrificial anode is required to maintain neutrality[1–13]. In contrast, paired electrocatalysis combines oxidative catalysis and reductive catalysis in an undivided cell, making it a more energy-efficient and economical electrolysis; however, this approach has remained underexploited[15–20]. In general, there are significant challenges associated with two-electrode reactions that are matched in terms of the reaction scale and rate required for convergent synthesis.

Molecular photoelectrochemistry, which combines photocatalysis and electrochemistry in one cell, has fascinated chemists recently[21–23]. In 2019, the Xu group reported a pioneering photoelectrochemical C−H alkylation of heteroarenes[24], Since then, photoelectrochemically driven diverse transformations, including C–H functionalization, C–X functionalization, decarboxylative coupling and Si–H functionalization, has begun to flourish (Fig. 1a)[16,20,24–53]. However, the more challenging C−C bond activation is relatively underdeveloped[40]. Alcohols are ubiquitous molecules throughout the chemical sciences, making them highly attractive starting materials for creating new C−C bonds[54]. However, C−C bonds in alcohols are kinetically inert and thermodynamically stable, and therefore, C−C

functionalization in alcohols typically relies on using strained molecules, i.e., ring strain provides the thermodynamic driving force[55]. Recently, ligand-to-metal charge transfer (LMCT) has been shown to be an efficient mechanism for directly generating alkoxy radicals from alcohols, thereby offering a new approach for synthetic transformations of ubiquitous alcohols[56,57]. However, reported systems usually involve Giese-type radical addition reactions, whereas redox-neutral methods remain largely underexplored[58–67]. Recently, the Lei group described the first electrophotochemical Ce-catalyzed ring-opening functionalization of cycloalkanols[40]. Considering recent advances in photoelectrochemistry and our ongoing interest in electrosynthesis[18,19,68–74], this study investigates the use of anodic photoelectrocatalysis to activate alcohols via ligand-to-metal charge transfer (LMCT) to yield alkyl radicals, which is coupled with electrophiles through cathodic transition metal catalysis to achieve C − C functionalization of unstrained alcohols (Fig. 1b). The feasibility of this method relies on the inhibition of alkoxy radical quenching via hydrogen atom transfer (HAT). Moreover, the rate of alkyl radical generation must match the relatively stable aryl-metal speciation. Accordingly, it is crucial to select suitable paired electrocatalysts that can provide sufficient rate-matching modulation of the system.

## Results and discussion

### Investigation of the reaction conditions

To probe the feasibility of our proposal, we initially chose 4-bromoacetophenone **1a** and 2-methyl-1-phenyl-2-propanol **2a** as substrates to examine the C − C arylation of alcohols in an undivided

[1]The Institute for Advanced Studies (IAS), Wuhan University, Wuhan, Hubei, P. R. China. [2]These authors contributed equally: Long Zou, Rui Sun. ✉e-mail: gci2011@whu.edu.cn

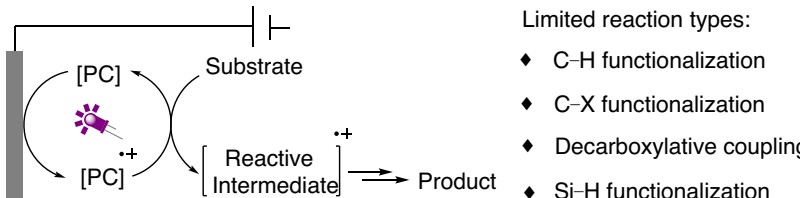

a) Photoelectrochemistry: underexploited in organic synthesis

Limited reaction types:

♦ C–H functionalization

♦ C–X functionalization

♦ Decarboxylative coupling

♦ Si–H functionalization

b) This work: Iron/nickel cocatalyzed C–C functionalization of unstrained alcohols

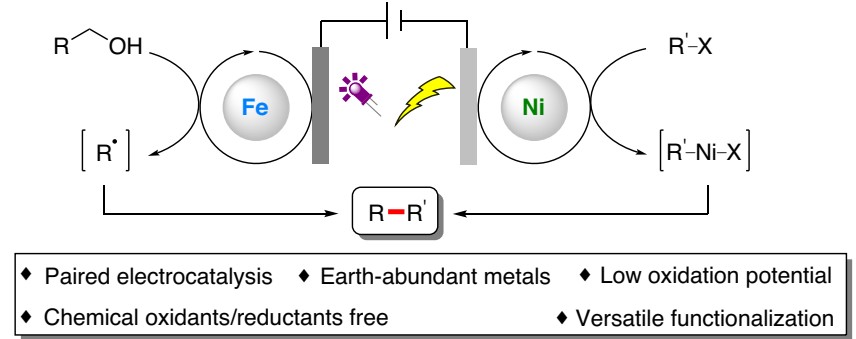

♦ Paired electrocatalysis    ♦ Earth-abundant metals    ♦ Low oxidation potential

♦ Chemical oxidants/reductants free                    ♦ Versatile functionalization

**Fig. 1 | Introduction. a** Photoelectrochemistry: underexploited in organic synthesis. **b** This work: Iron/nickel cocatalyzed C – C functionalization of unstrained alcohols.

## Table 1 | Optimization of the reaction conditions [a]

GF (+) | GF (-), FeCl₃ (10 mol%)
NiCl₂·DME (10 mol%), dtbbpy (10 mol%)
TBAPF₆, Cs₂CO₃, MeCN, 4 mA, 8 h
undivided cell, 20 W 398 - 400 nm

| Entry | Variation from standard conditions | Yield (%)[b] |
|---|---|---|
| 1 | none | 89 (85) |
| 2 | CeCl₃ instead of FeCl₃ | n.d. |
| 3 | CuCl₂ instead of FeCl₃ | n.d. |
| 4 | CoCl₂ instead of NiCl₂·DME | n.d. |
| 5 | phenanthroline instead of dtbbpy | 15 |
| 6 | tpy instead of dtbbpy | 6 |
| 7 | bpy instead of dtbbpy | 36 |
| 8 | LiClO₄ instead of TBAPF₆ | 4 |
| 9 | DMF instead of MeCN | 8 |
| 10 | THF instead of MeCN | 9 |
| 11 | 10 W 398–400 nm | 36 |
| 12 | 30 W 398–400 nm | 63 |
| 13 | I = 6 mA | 70 |
| 14 | w/o Cs₂CO₃ | n.d. |
| 15 | w/o FeCl₃ or NiCl₂·DME | n.d. |
| 16 | w/o electricity or light | n.d. |

ᵃReaction conditions: **1a** (0.3 mmol), **2a** (3.0 equiv.), FeCl₃ (10 mol%), NiCl₂·DME (10 mol%), dtbbpy (10 mol%), TBAPF₆ (1.0 equiv.), Cs₂CO₃ (2.0 equiv.), anhydrous MeCN (6.0 mL), 4 mA, 8 h, 20 W
398–400 nm, 25 °C, argon, graphite felt (GF) as electrodes, undivided cell.
ᵇGC yields using biphenyl as an internal standard, isolated yields in parentheses. tpy = 2,6-bis(2-pyridyl)pyridine. bpy = 2,2'-bipyridine. dtbbpy = 4,4'-di-tert-butyl-2-2'-bipyridine. w/o=without. n.d.=not
detected.

cell. Evaluations of key parameters (e.g., catalyst, ligand, solvent, electrode, electrolyte; Table 1) revealed that the desired dehydroxymethylative arylation product **1** can be isolated in 85% yield using commercially available FeCl₃ and NiCl₂·DME as the paired catalysts under irradiation by purple LEDs (entry 1). Notably, the desired product was not detected when CeCl₃ was employed as the anodic

catalyst, despite its ability to activate alcohol via ligand-to-metal charge transfer (LMCT) (entry 2). The low cell voltage indicates that a short circuit might arise due to CeCl₃ under these conditions, which would limit the conversion of both starting materials. Similarly, CuCl₂ was also inefficient in this process, likely because it can be easily deposited on the surface of the cathode, even at low reduction

potential (entry 3). Cathodic catalysts (e.g., CoCl$_2$) that have much lower reactivities than NiCl$_2$·DME failed to afford the desired product (entry 4). These results highlight the profound impacts of the anodic and cathodic catalysts. When other ligands, such as phenanthroline, 2,6-bis(2-pyridyl)pyridine (tpy), or 2,2'-bipyridine (bpy) were applied instead of 4,4'-di-tert-butyl-2-2'-bipyridine (dtbbpy), the yield of the desired product decreased significantly (entries 5-7). Lithium salt, which is a commonly used electrolyte, also leads to a low yield (entry 8). In this case, coordination between the Li-ion and alcohols might significantly weaken the interaction between the alcohol and the anodic catalyst. The solvent can serve as a hydrogen atom donor in radical processes, e.g., N,N-Dimethylformamide and Tetrahydrofuran, thereby dramatically inhibiting the desired reaction (entries 9-10). The chosen solvent should suppress alkoxy radical quenching and accelerate the β-scission of the primary alkoxy radicals. Therefore, acetonitrile is an appropriate solvent for this reactivity because it is sufficiently resistant to hydrogen atom transfer (HAT) processes involving alkoxy radicals. To further promote the reaction efficiency, modulation of the light source and current intensity was applied to tune the anodic/cathodic half-reactions. Indeed, such a rate-matched regulation has a great influence on the reaction selectivity (entries 11-13). Including Cs$_2$CO$_3$ in the reaction system is crucial because it can neutralize the HBr generated in situ to maintain the neutrality of the reaction (entry 14). Finally, control experiments confirmed that the paired catalysts, electrical current, and light irradiation are all essential, i.e., the desired product was not detected when the reaction was performed in their absence (entries 15-16).

## Scope of substrates

The generalizability of the reaction in terms of the alcohol component was investigated. As shown in Figure 2, a wide range of primary alcohols could be used as operationally simple carbon pronucleophiles in the cross-coupling reactions with aryl bromides (**1**–**12**). Alcohols containing electron-poor or electron-rich arenes were viable in this transformation, affording the desired products (**2**–**6**) in 62%-77% yields. Various alkyl substitutions (linear or cyclic) at the α-position of the hydroxyl group were suitable for this protocol (**8**–**12**). Notably, primary alcohols with heteroatom and alkene functionalities were well tolerated under the standard reaction conditions, delivering the corresponding target products **9**–**12** in moderate yields. In addition, this reaction was effective for secondary and tertiary alcohols, affording the expected **1** in good or excellent yields.

Subsequently, the reaction scope of aryl bromides was investigated (Fig. 2, **13**–**30**). A series of aryl bromides bearing electron-withdrawing groups that can be reduced under electroreductive conditions (e.g., nitrile, methylsulfonyl, ester, and ketone) were well tolerated in this reaction, furnishing the corresponding products (**13**–**18**) in 68%-88% yields. Moreover, various electron-donating groups (e.g., methoxyl, methylthiol, benzofuran, benzothiophene, and thiophene) that are usually sensitive to oxidative conditions were compatible with this protocol as well (**24**–**28**). Furthermore, valuable pyridine and isoquinoline derivatives **29** and **30** were delivered in excellent yields when electro-deficient heteroaromatic bromides were employed. Notably, Minisci product from alkyl radical to heteroarenes was not observed in this transformation.

The developed approach can also facilitate the arylation of remote sites on ketones by promoting the ring-opening of unstrained cycloalkanols. For example, unstrained cyclohexanols (ring strain energy ≈ 0.1 kcal/mol) were viable in this C − C cleavage/arylation reaction[26]. A wide range of aryl bromides bearing a variety of substituents, such as ketones, esters, sulfones, aldehydes, fluorine, chlorine, and methylthiol, could be coupled with cyclohexanol to afford the corresponding ketone products (**31**–**48**) in 33%-90% yields. Of note, aryl bromides bearing amino groups can be tolerated in this reaction (**40**–**42**), the corresponding yields decreased along with the number of substitutes decreases on the amino group. Electron-poor heteroarenes (e.g., isoquinoline) and electron-rich heteroarenes (e.g., carbazole) survived well during electrolysis (**43**–**44**). In addition, aryl or alkyl groups adjacent to the reactive hydroxyl group in cyclohexanol were well-tolerated, furnishing the corresponding products (**45**–**46**) in 75%-80% yields. Heterocyclic alcohols, e.g., 4-methyltetrahydro-2H-pyran-4-ol, were amenable to this protocol as well, giving product **47** in 72% yield. Less-strained cycloalkanols, e.g., 1-methylcyclopentan-1-ol, also served as suitable reaction partners and gave the corresponding product **48** in 68% yield.

The developed strategy provides a general photoelectrocatalytic platform for C − C functionalization of alcohols. As shown in Figure 3a, C − C alkenylation of alcohols was achieved using β-bromostyrene as the coupling partner, giving the corresponding product **49** in 76% yield. Chlorobenzene, which has a C−Cl bond (much stronger than a C−Br bond), was also a compatible reaction substrate, affording the corresponding product **1** in moderate yield. In addition, the photoelectrocatalytic strategy can be scalable with high efficiency. For example, the desired C(sp$^2$)−C(sp$^3$) coupling product **1** was isolated with 88% yield in a gram-scale synthesis (1.11 g, Fig. 3b). Furthermore, the developed method was also applicable to late-stage modifications of various natural products and pharmaceutical derivatives (Fig. 3c). Various aryl bromides derived from naproxen, thiomorpholine, menthol, geraniol, flurbiprofen, and diacetonefructose were coupled with alcohols to furnish the C − C arylation products (**50**–**57**) in 46%-80% yields. Therefore, this approach provides easy access to building blocks for pharmaceutical drug discovery efforts. Nevertheless, long-chain primary alcohol, such as 1-hexanol, was not compatible with this protocol, presumably due to the unfavorable β-scission of the primary alkoxyl radical. In addition, substrates with strong coordinating groups, which might coordinate with FeCl$_3$ catalyst and deactivate the catalyst completely (e.g., 4-bromophenol, 4-bromothiophenol, 2-phenylethanethiol), were also not compatible in this protocol. Furthermore, no desired diketone was detected when benzoyl chloride was employed to react with 1-ethylcyclohexan-1-ol in this reaction (Fig. 3d).

## Mechanistic studies

A series of experiments were performed to gain mechanistic insights. First, the desired product **1** was obtained in 68% yield, along with benzaldehyde in 25% yield, when 1,2-diphenylethanol (**2b**) was used under the standard conditions (Fig. 4a). The model reaction was completely inhibited if the radical acceptor, methyl 2-((phenylsulfonyl)methyl)acrylate, was added to the standard reaction, and the benzylic radical addition/substitution product **59** was obtained in 7% yield (Fig. 4b). These results revealed that a β-scission process was involved in this reaction; specifically, the α-hydroxy C − C bond of the alcohol was cleaved to yield the corresponding alkyl radical. In addition, the desired product was not detected when the reaction was conducted using benzylcetyldimethylammonium chloride (HDBAC) or tetrabutylammonium bromide (TBAB) as the additive in the absence of FeCl$_3$ (Fig. 4c-d) under the standard conditions. The potential selected for these experiments was sufficient to directly oxidize chloride and bromide anions. Moreover, the corresponding benzyl chloride or benzyl bromide, which can be generated through halogenation of the benzyl radical with chlorine or bromine radicals generated in situ, was not detected by gas chromatography-mass spectrometry (GC-MS) analysis after the standard reaction (Fig. 4e). These results indicate that chlorine or bromine radicals can be excluded as key active intermediates for the activation of alcohols[75].

Further investigation showed that the plot of initial reaction rate versus current intensity is linear, indicating that the step of electrode reaction is the rate-liming (Fig. 5a). Next, the electrode voltage for the model reaction was monitored during electrolysis. The anode/cathode potential was kept around 0.3 V and −0.70 V vs. Ag/AgCl (Fig. 5b),

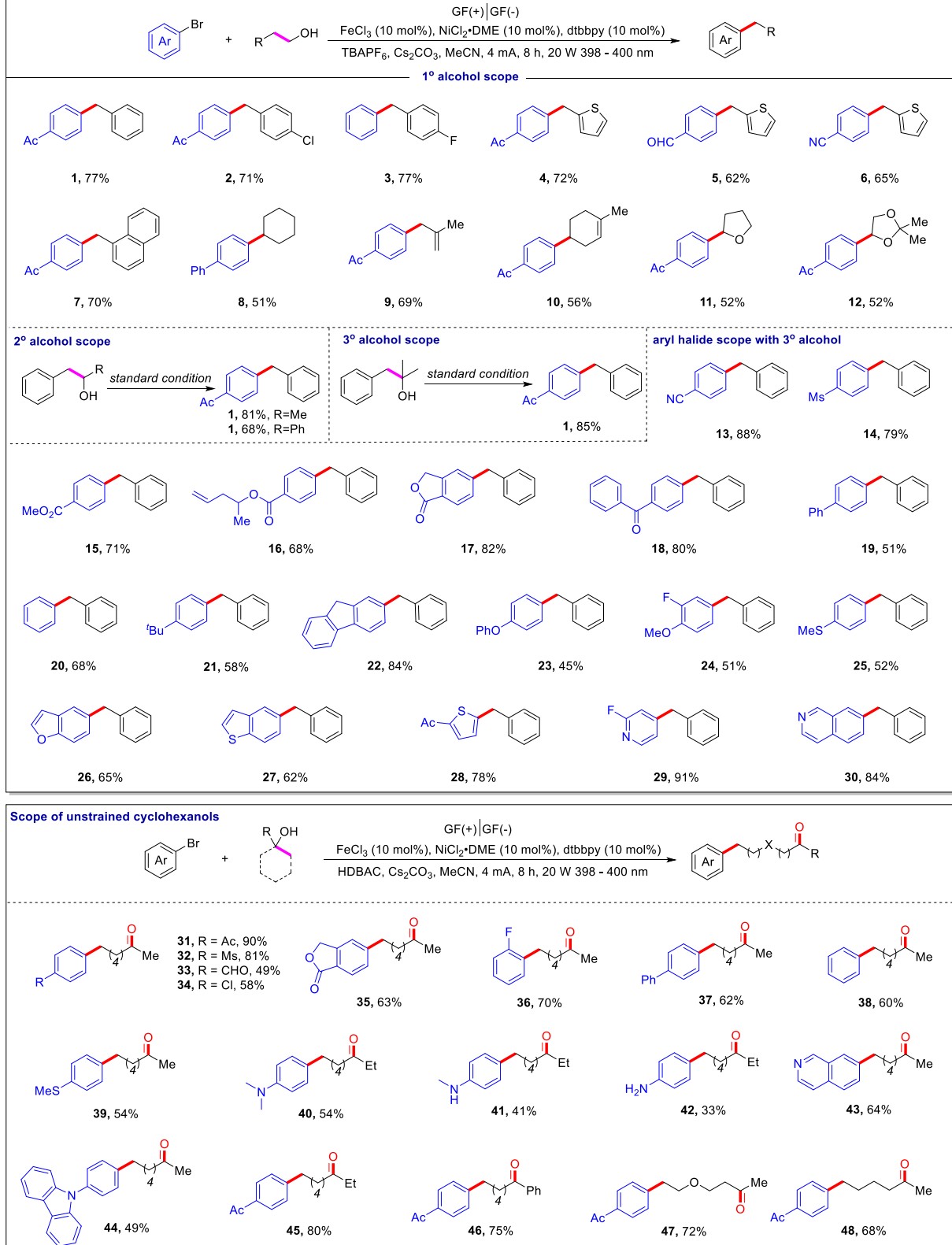

**Fig. 2 | Substrate scope for photoelectrochemical C−C functionalization of alcohols.** For details, see the Supplementary Information.

respectively, which is consistent with catalytic cycle of Fe$^{II}$/Fe$^{III}$ and ArNi$^{III}$/ArNi$^{II}$ (e.g., for ArNi$^{III}$(Mebpy)Br$_2$, Ar = 4-CF$_3$C$_6$H$_4$, Calc. $E_{red}$ ≈ −0.5 V vs. Ag/AgNO$_3$)[76].

On the basis of the aforementioned results and previous reports[18,19,58–67], a possible reaction pathway is depicted in Fig. 6. Alkoxy radical (**D**) is generated via photoelectrocatalytic activation of alcohol with iron at the anode, which is followed by β-scission to yield alkyl radical **E**. Meanwhile, organohalide (electrophile) reacts with low valent Ni$^{I}$ species formed in situ and further undergoes cathodic reduction, affording R′-Ni$^{II}$ species (**H**). Afterward, alkyl radical **E** is

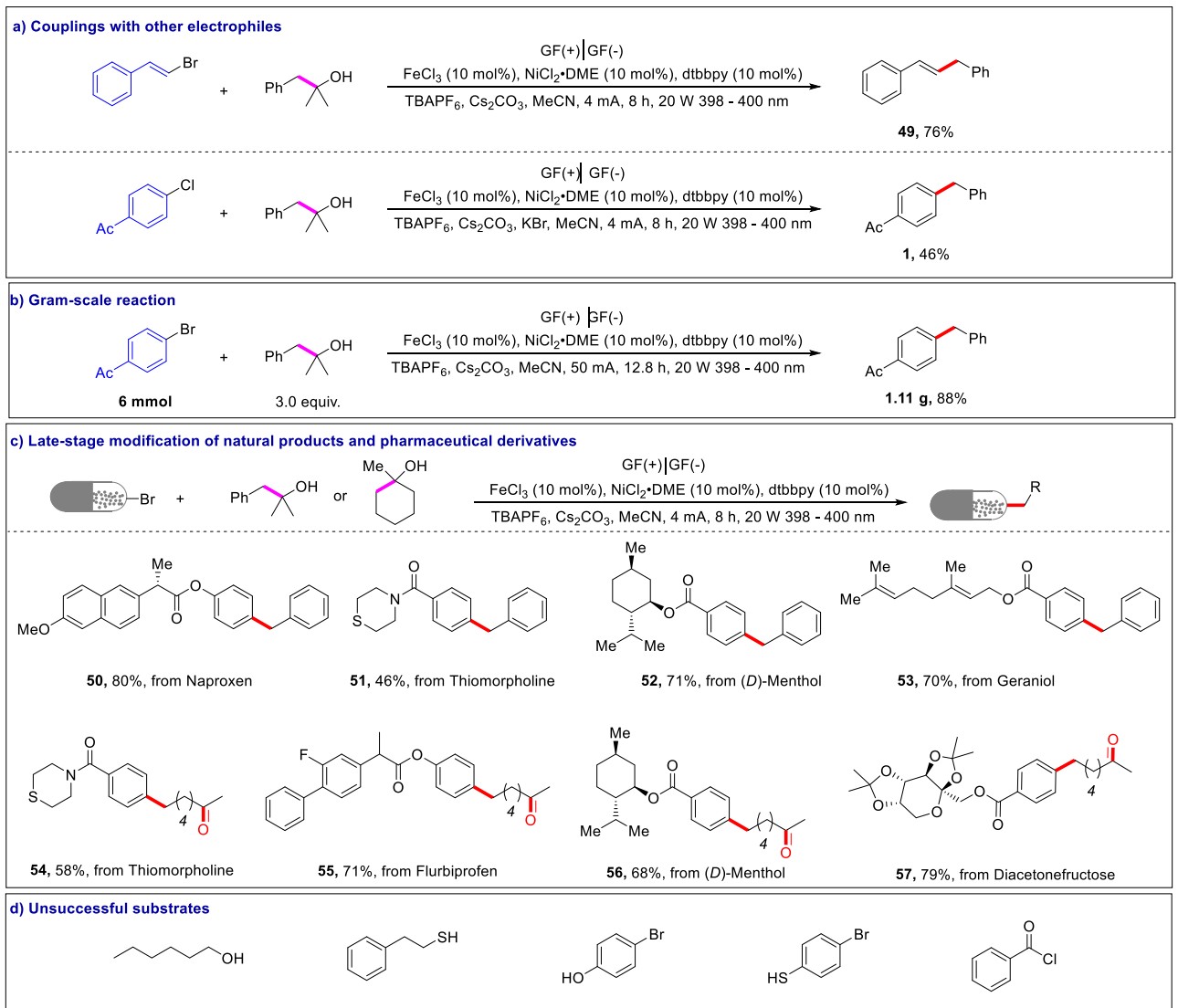

**Fig. 3 | Substrate scope for photoelectrochemical C−C functionalization of alcohols. a** Couplings with other electrophiles. **b** Gram-scale reaction. **c** Late-stage modification of natural products and pharmaceutical derivatives. **d** Unsuccessful substrates. For details, see the Supplementary Information.

trapped by R′-Ni$^{II}$ species **H** to deliver the high-valent R-Ni$^{III}$-R′ species (**I**). Further reductive elimination would furnish the dehydroxymethylative functionalization product **J** and recycle the Ni$^{I}$ catalyst **F**. Notably, **E** are typically transient radicals whose lifetime is less than a millisecond[77,78]. In comparison, aryl-Ni$^{II}$ species are relatively stable, which can be prepared and isolated as pure compounds, as in previous reports[79]. The high reaction selectivity for cross-coupling might be similar to the persistent radical effect[80].

In conclusion, a method for the photoelectrochemical C − C functionalization of alcohols was developed, in which paired oxidative and reductive catalysis occurs in a single cell without interference. A wide range of alcohols (primary, secondary, tertiary, or unstrained cyclic alcohols) were effectively utilized in this protocol under mild electrochemical conditions. This reaction exhibits broad functional group compatibility and can be used to generate a diverse series of valuable molecules (more than 50 examples described herein). Mechanistic studies indicated that photoelectrochemical iron catalysis occurs at the anode to activate the alcohol, while nickel catalysis at the cathode activates electrophiles. This system avoids the use of stochiometric chemical reductants in the reduction process or oxidants in the oxidative activation of alcohols. The developed protocol offers a green synthetic method that can inspire

further applications to support diverse transformations in chemical synthesis.

## Methods
### Representative procedure for the synthesis of compounds (1−30, 50−53)

In an oven-dried two-necked cell (20 mL) equipped with a Teflon-coated magnetic stir bar and two graphite felt electrodes (15 mm × 15 mm × 2.5 mm), Cesium carbonate (196 mg, 0.6 mmol, 2.0 equiv), TBAPF$_6$ (116.2 mg, 0.3 mmol, 1.0 equiv.) and FeCl$_3$ (4.8 mg, 10 mol%) were added in a glovebox. The reaction cell was sealed and moved out from the glovebox. Afterward, a pre-catalyst solution (was prepared by a mix of NiCl$_2$·DME (6.6 mg, 10 mol%), 4,4′-di-tert-butyl-2,2′-bipyridine (8.1 mg, 10 mol%) in anhydrous MeCN (6.0 mL) under argon atmosphere, and was stirred for 5 minutes.), aryl bromides (0.3 mmol, 1.0 equiv.) and alcohols (0.9 mmol, 3.0 equiv.) were added to the reaction cell via a syringe. The reaction mixture was pre-stirred for 5 minutes and was electrolyzed at a constant current of 4 mA under irradiation by a 20 W purple LED lamp (0.3 cm away, with a cooling fan to keep the reaction temperature at 25 °C) for 8 h. After the reaction, the reaction mixture was concentrated (the residual product on electrodes were rinsed with ethyl acetate), and purified

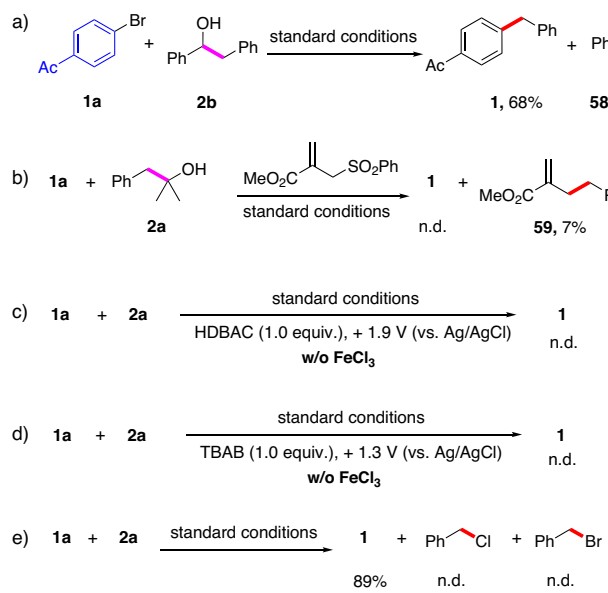

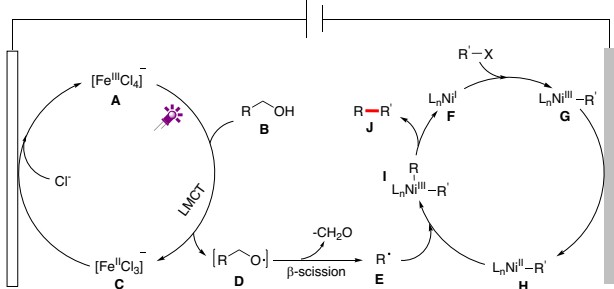

**Fig. 6 | Proposed mechanism.** Proposed photoelectrochemical C–C functionalization of alcohols.

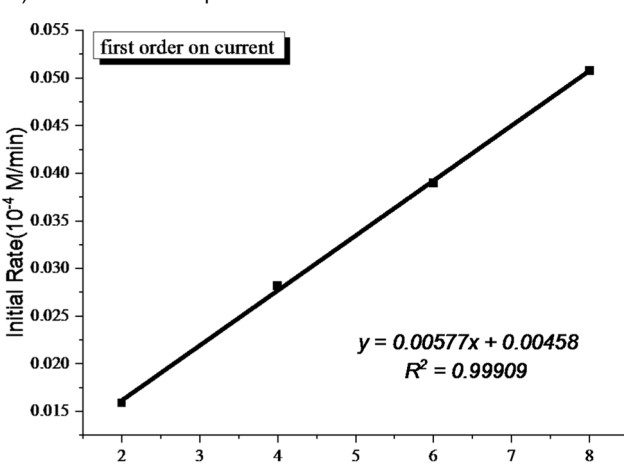

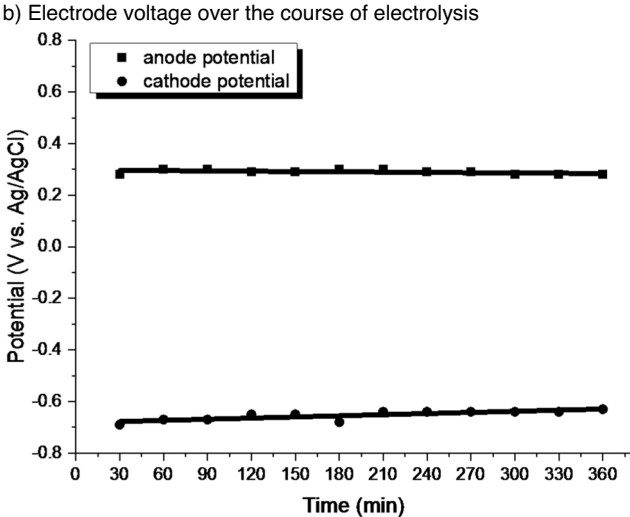

**Fig. 4 | Mechanistic studies. a** The reaction of **1a** with **2b**. **b** Radical trapping experiment. **c, d** Constant Voltage Electrolysis. **e** The reaction of **1a** with **2a**.

a) First-order rate dependence on current

$y = 0.00577x + 0.00458$
$R^2 = 0.99909$

b) Electrode voltage over the course of electrolysis

**Fig. 5 | Mechanistic studies. a** Plot showing first-order rate dependence on current. **b** Electrode voltage over the course of electrolysis.

by column chromatography (eluted with ethyl acetate/petroleum ether) to afford the pure product.

### Representative procedure for the synthesis of compounds (31–48, 54–57)

In an oven-dried two-necked cell (20 mL) equipped with a Teflon-coated magnetic stir bar and two graphite felt electrodes (15 mm × 15 mm × 2.5 mm), Cesium carbonate (196 mg, 0.6 mmol, 2.0 equiv.), HDBAC (Benzyldimethylhexadecylammonium chloride) (118.8 mg, 0.3 mmol, 1.0 equiv.) and FeCl₃ (4.8 mg, 10 mol%) were added in a glovebox. The reaction cell was sealed and moved out from the glovebox. Afterward, a pre-catalyst solution (was prepared by a mix of NiCl₂·DME (6.6 mg, 10 mol%), 4,4'-di-tert-butyl-2,2'-bipyridine (8.1 mg, 10 mol%) in anhydrous MeCN (6.0 mL) under argon atmosphere, and was stirred for 5 minutes.), aryl bromides (0.3 mmol, 1.0 equiv.) and alcohols (0.9 mmol, 3.0 equiv.) were added to the reaction cell via a syringe. The reaction mixture was pre-stirred for 5 minutes and was electrolyzed at a constant current of 4 mA under irradiation by a 20 W purple LED lamp (0.3 cm away, with a cooling fan to keep the reaction temperature at 25 °C) for 8 h. After the reaction, the reaction mixture was concentrated (the residual product on electrodes were rinsed with ethyl acetate), and purified by column chromatography (eluted with ethyl acetate/petroleum ether) to afford the pure product.

## Data availability

Data relating to the characterization data of materials and products, general methods, optimization studies, experimental procedures, mechanistic studies and NMR spectra are available in the Supplementary Information. All data are also available from the corresponding author upon request.

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

## Acknowledgements

This work was supported by the Fundamental Research Funds for the Central Universities (2042024kf1040), the National Natural Science Foundation of China (No. 22271227), the National Key R&D Program of China (2021YFA1500100), and Wuhan University.

## Author contributions

Q.L. conceived and directed the project. L.Z. and R.S. conducted most of the experimental studies. Y.T., X.W. and X.Z. supported performance of synthetic experiments. Q.L. wrote the manuscript. All authors discussed the results, analyzed the data, and prepared the manuscript.

## Competing interests

The authors declare no competing interests.
