## [Peer Review File · Nature Communications]

Photoelectrochemical Fe/Ni Cocatalyzed C–C Functionalization of AlcoholsReviewers' Comments:

Reviewer #1:

Remarks to the Author:

This paper describes a method for the photoelectrochemical C–C functionalization of alcohols based on a paired oxidative and reductive catalysis. The reaction system is very simply assembled and efficient, fully utilizing the capabilities of paired electrolysis. The products obtained are those that require more complex steps to obtain by ordinary chemical methods. The system is also likely to be widely useful in modifying natural products and existing pharmaceutical raw materials. In order to envision further chemical development based on this reaction system in present method, it is necessary to describe the properties of this system in more detail.

For this purpose, it is necessary to analyze the function of Fe and Ni catalysis, the lifetime of the complex formed with R and R', respectively, and the formation rate of each intermediate at both poles. For this purpose, the following verification results are required.

(1) Numerous examples are shown in this paper, and it is understood that the system is highly versatile. On the other hand, what substrates are difficult to apply to this reaction system or whose yields are significantly reduced? For example, can substrates with thiol or amino groups yield high levels of product? What other factors may inactivate Fe and Ni catalysis? Such scope and limitation should be shown.

(2) In paired electrolysis, the efficiency (reaction rate) of active intermediates at both electrodes, the lifetime of unstable intermediates, and the reaction rate between intermediates generated from both electrodes are important factors to determine the overall reaction efficiency. Analysis of these factors is sought. In other words, what is the lifetime of the anode and cathode complexes with R and R', respectively, and can a homo-coupling between R radicals occur as a side reaction? Based on these comprehensive findings, the reaction system should be evaluated.

These clarifications make this paper of high academic value.

Reviewer #2:

Remarks to the Author:

The manuscript presents a photoelectrochemical strategy for the carbon-carbon bond cleaving cross-coupling of tertiary alcohols with aryl halides. However, the study's scope appears constrained as it requires specific homobenzylic or cyclic tertiary alcohols. Not only is the carbon-carbon cleavage of these alcohols through photochemical means well established, but the utility of the resulting products is also highly questionable. In many cases, these compounds can be synthesized more efficiently from alternative, more accessible starting materials, or they lack practical applications. For instance, benzyl groups can be easily introduced through benzylic halides, offering a more convenient route than through tertiary alcohols. Although photoelectrochemical methods are indeed valuable and cutting-edge, the narrow scope of this study, along with the well-established nature of cross-coupling reactions, limits its overall significance.

Reviewer #3:

Remarks to the Author:

Alcohols are ubiquitous molecules throughout the chemical sciences, making them highly attractive starting materials for creating new C–C bonds. However, C–C bonds in alcohols are kinetically inert and thermodynamically stable, and therefore, C–C functionalization in alcohols typically relies on using strained molecules. In this manuscript, the Lu group developed an electrophotocatalysis based on earth-abundant iron/nickel, providing new avenues for C–C arylation/alkenylation of alcohols. Of

note, electrophotochemistry, which integrates the merits of both electrochemistry and photocatalysis while overcoming their flaws, has recently fascinated organic chemists. However, the development of synergistic use of electrophotochemistry with paired electrocatalysis is just in its infancy (only around five papers). Accordingly, there is no doubt that this strategy would be a powerful platform for more transformations of ubiquitous alcohols, providing flexibility not possible with single photoredox catalysis. Importantly, this work has the features of broad substrate scope (i.e., primary, secondary, tertiary, or unstrained cyclic alcohols), low oxidation potential applied (~ 0.30 V vs. Ag/AgCl), late-stage modification of complex molecules and insightfully mechanism investigation. Overall, this is a sound work that has been carefully and competently carried out, and the impact of the work is worthy of publication in Nature Communications.

Some suggestions are as follows:

1. Figure 5b: how were the potential of the anode and the cathode simultaneously monitored?
2. Scale-up run: the current was multiplied 12.5-fold. Was the size of the electrodes also increased accordingly?
3. Further application: the elegant strategy provides a powerful platform for C–C functionalization of ubiquitous alcohols, which has been proved in Figure 3a in the manuscript. How about other electrophiles, such as acyl chlorides? It would be an interesting pathway for the construction of useful and valuable diketones.

We would like to deliver our sincere thanks to the reviewers who provided helpful and positive comments on this manuscript. Taking into consideration the suggestions from the reviewers, the manuscript has been carefully revised according to the comments. The following list is our point-to-point response to Reviewers' comments.

Response to the comments from Reviewer #1 (Remarks to the Author):

Comments:

Reviewer 1: This paper describes a method for the photoelectrochemical C–C functionalization of alcohols based on a paired oxidative and reductive catalysis. The reaction system is very simply assembled and efficient, fully utilizing the capabilities of paired electrolysis. The products obtained are those that require more complex steps to obtain by ordinary chemical methods. The system is also likely to be widely useful in modifying natural products and existing pharmaceutical raw materials. In order to envision further chemical development based on this reaction system in present method, it is necessary to describe the properties of this system in more detail.

For this purpose, it is necessary to analyze the function of Fe and Ni catalysis, the lifetime of the complex formed with R and R', respectively, and the formation rate of each intermediate at both poles. For this purpose, the following verification results are required.

Response:

We thank the reviewer for the positive recommendation.

Comments:

(1) Numerous examples are shown in this paper, and it is understood that the system is highly versatile. On the other hand, what substrates are difficult to apply to this reaction system or whose yields are significantly reduced? For example, can substrates with thiol or amino groups yield high levels of product? What other factors may inactivate Fe and Ni catalysis? Such scope and limitation should be shown.

Response:

Long-chain primary alcohol, such as 1-hexanol, was not compatible in this protocol, presumably due to the unfavorable β -scission of the primary alkoxy radical. In addition, substrates with strong coordinating groups, which might coordinate with FeCl_3 catalyst and deactivate the catalyst completely (e.g., 4-bromophenol, 4-bromothiophenol, 2-phenylethanethiol), were also not compatible in this protocol. Furthermore, aryl bromides bearing amino groups can be tolerated in this reaction. Of note, the corresponding yields decreased along with the number of substitutes decreases on the amino group (see below). These results have been added to the revised manuscript.

Comments:

(2) In paired electrolysis, the efficiency (reaction rate) of active intermediates at both electrodes, the lifetime of unstable intermediates, and the reaction rate between intermediates generated from both electrodes are important factors to determine the overall reaction efficiency. Analysis of these factors is sought. In other words, what is the lifetime of the anode and cathode complexes with R and R', respectively, and can a homo-coupling between R radicals occur as a side reaction? Based on these comprehensive findings, the reaction system should be evaluated.

Response:

Alkyl radicals generated in this reaction are typically transient species with a lifetime of less than a millisecond (*Acc. Chem. Res.* **1976**, *9*, 13; *Chem. Sci.* **2021**, *12*, 13158.). In comparison, aryl- Ni^{II} species are relatively stable, which can be prepared and isolated as pure compound, as in previous reports (e.g., *J. Am. Chem. Soc.* **2018**, *140*, 12200; *ACS Catal.* **2024**, *14*, 6897.). In this work, the key to the success of this protocol is that the rate of alkyl radical generation must match the relatively stable aryl-metal speciation. To achieve the rate-matched model of paired electrolysis, the ligand-to-metal charge transfer process associated with the anodic half-reactions is separately modulated by the light source while the cathodic half-reactions is tuned by dialing in the current. Electrophotocatalytic reaction selectivity benefits from such energy-input tuning, as evidenced by the different results after changing the light source and current intensity (entries 11-13, Table 1 in the manuscript).

In addition, no obvious homocoupling of alkyl radicals was observed under standard condition by GC-MS analysis (as shown in the following figure). The high reaction selectivity for cross-coupling might be similar to persistent radical effect (*Angew. Chem. Int. Ed.* **2020**, *59*, 74.): the more reactive transient radical is preferably consumed at the initial stage of the reaction by homocoupling or disproportionation, whereas the longer-lived "persistent" species can accumulate in solution during that period. As a result, although the two radicals are generated at equal rates, they are present in solution at significantly different concentrations and cross-coupling becomes the dominant process.

Supplementary Figure 28. The reaction mixture was analyzed by GC-MS.

Comments:

These clarifications make this paper of high academic value.

Response:

We would like to deliver our sincere thanks to the reviewer who provided helpful and positive comments on this manuscript, and the manuscript has been carefully revised according to the comments.

Response to the comments from Reviewer #2 (Remarks to the Author):

Comments:

Reviewer 2: The manuscript presents a photoelectrochemical strategy for the carbon-carbon bond cleaving cross-coupling of tertiary alcohols with aryl halides. However, the study's scope appears constrained as it requires specific homobenzylic or cyclic tertiary alcohols. Not only is the carbon-carbon cleavage of these alcohols through photochemical means well established, but the utility of the resulting products is also highly questionable. In many cases, these compounds can be synthesized more efficiently from alternative, more accessible starting materials, or they lack practical applications. For instance, benzyl groups can be easily introduced through benzylic halides, offering a more convenient route than through tertiary alcohols. Although photoelectrochemical methods are indeed valuable and cutting-edge, the narrow scope of this study, along with the well-established nature of cross-coupling reactions, limits its overall significance.

Response:

We believe Reviewer 2 made some misunderstanding: 1) Most of the alcohols employed in this reaction are primary alcohols, only one chain tertiary alcohol is employed in this reaction, that is 2-methyl-1-phenyl-2-propanol (**2a**) in the standard reaction. 2) Not only aryl ethanol and unstrained cycloalkanols are compatible with this protocol, but also a series of primary alcohols with alkyl substituents are well tolerated (e.g., products **8-12** in Figure 2) in this reaction.

We respectfully disagree with the reviewer. The reasons are as follows:

1) Importance: Organic electrocatalysis offers a powerful and sustainable alternative to conventional chemical manufacturing, which has been demonstrated as an important topic by recent high-profile publications: *Nature* **2023**, 623, 745; *Nature* **2023**, 623, 71; *Science* **2023**, 379, 1036; *Science* **2023**, 380, 81; *Science* **2024**, 384, 113.

2) Novelty: Just as reviewer 3 commented: ‘...the development of synergistic use of electrophotocatalysis with paired electrocatalysis is just in its infancy (only around five papers).’ Herein, a method for the photoelectrochemical C–C functionalization of various alcohols via paired electrocatalysis was developed.

3) General Platform: A general strategy for C–C functionalization of alcohols is developed. A variety of alcohols (i.e., primary, secondary, tertiary, or unstrained cyclic alcohols) can be activated via photoelectrocatalysis coupled with versatile electrophiles (e.g., aryl bromides, aryl chloride and β -bromostyrene).

4) Excellent Functional Group Tolerance: Photoelectrocatalytic activation of alcohols with ultra-low oxidation potential (~ 0.30 V vs. Ag/AgCl) is achieved, thus enabling compatibility with a wide range of functional groups. Functional groups that are typically sensitive to oxidative conditions, such as methoxyl, methylthiol, benzofuran, benzothiophene and thiophene, were well tolerated (more than 50 examples).

5) Utility and Application: The resulting products, diarylmethanes, ketones and olefins, are common structural features in many natural products, pharmaceuticals, organic materials and agrochemicals etc. In addition, the developed method was further applied into late-stage modifications of structurally complex natural products and pharmaceutical derivatives (Figure 3c in the manuscript), providing easy access to building blocks for pharmaceutical drug discovery efforts.

6) Step Economy: Compared with alkyl halides (e.g., benzylic halides are usually prepared from the corresponding benzylic alcohols, resulting in arduous multistep synthetic sequences), alcohols are the most widely available alkyl fragment (ref 54: *Nature* **2021**, 598, 451.). Therefore, the use of alcohols as C(sp³) fragments in cross-coupling reactions is highly desirable.

Overall, we believe that this manuscript not only provides a new methodology for the longstanding synthetic challenge in C–C functionalization of alcohols, but also should inspire advances in other photoelectrocatalytic reactions based on the concept outlined here.

Response to the comments from Reviewer #3 (Remarks to the Author):

Comments:

Reviewer 3: Alcohols are ubiquitous molecules throughout the chemical sciences, making them highly attractive starting materials for creating new C–C bonds. However, C–C bonds in alcohols are kinetically inert and thermodynamically stable, and therefore, C–C functionalization in alcohols typically relies on using strained molecules. In this manuscript, the Lu group developed an electrophotocatalysis based on earth-abundant iron/nickel, providing new avenues for C–C arylation/alkenylation of alcohols. Of note, electrophotochemistry, which integrates the merits of both electrochemistry and photocatalysis while overcoming their flaws, has recently fascinated organic chemists. However, the development of synergistic use of electrophotochemistry with paired electrocatalysis is just in its infancy (only around five papers). Accordingly, there is no doubt that this strategy would be a powerful platform for more transformations of ubiquitous alcohols, providing flexibility not possible with single photoredox catalysis. Importantly, this work has the features of broad substrate scope (i.e., primary, secondary, tertiary, or unstrained cyclic alcohols), low oxidation potential applied (~0.30 V vs. Ag/AgCl), late-stage modification of complex molecules and insightfully mechanism investigation. Overall, this is a sound work that has been carefully and competently carried out, and the impact of the work is worthy of publication in *Nature Communications*.

Response:

We thank the reviewer for the positive recommendation.

Comments:

Some suggestions are as follows:

1. Figure 5b: how were the potential of the anode and the cathode simultaneously monitored?

Response:

The electrodes and reference electrode (Ag/AgCl as a reference electrode) were inserted into the reaction mixture together during the electrolysis. A multimeter was used to monitor electrode potential between anode/cathode and reference electrode respectively.

Comments:

2: Scale-up run: the current was multiplied 12.5-fold. Was the size of the electrodes also increased accordingly?

Response:

For the scale-up reaction, the size of the electrodes increased from 1.5 cm*1.5 cm to 3.5 cm*4.0 cm. The graphical guide for scale-up reaction is presented in the SI (section 2.5).

Comments:

3: Further application: the elegant strategy provides a powerful platform for C–C functionalization of ubiquitous alcohols, which has been proved in Figure 3a in the manuscript. How about other electrophiles, such as acyl chlorides? It would be an interesting pathway for the construction of useful and valuable diketones.

Response:

Other electrophiles, such as β -bromostyrene and chlorobenzene, were tolerated in this reaction (Figure 3a in the manuscript). However, no desired diketone was detected when benzoyl chloride was employed to react with 1-ethylcyclohexan-1-ol in this reaction.

Reviewers' Comments:

Reviewer #1:

Remarks to the Author:

This revised manuscript appropriately addresses the comments of the reviewers. It has a high academic impact and is acceptable in its present form.

Reviewer #3:

Remarks to the Author:

The paper's author responded very well to my comments. I have no further questions and recommend accepting it as it is.

We are very pleased to hear that our manuscript, titled “Photoelectrochemical Fe/Ni Cocatalyzed C–C Functionalization of Alcohols”, is, in principle, accepted in *Nature Communications*. We thank you for your kind consideration, and also would like to deliver our sincere thanks to the reviewers who provided helpful and positive comments on this manuscript. The manuscript has been carefully revised according to the editorial requests.

Response to the comments from Reviewer #1 (Remarks to the Author):

Comments:

Reviewer 1: This revised manuscript appropriately addresses the comments of the reviewers. It has a high academic impact and is acceptable in its present form.

Response:

Thanks for the support.

Response to the comments from Reviewer #3 (Remarks to the Author):

Comments:

Reviewer 1: The paper's author responded very well to my comments. I have no further questions and recommend accepting it as it is.

Response:

Thanks for the support.